# Genome-Wide Identification and Characterization of Tomato Fatty Acid β-Oxidase Family Genes *KAT* and *MFP*

**DOI:** 10.3390/ijms25042273

**Published:** 2024-02-14

**Authors:** Long Li, Zesheng Liu, Xuejuan Pan, Kangding Yao, Yuanhui Wang, Tingyue Yang, Guohong Huang, Weibiao Liao, Chunlei Wang

**Affiliations:** College of Horticulture, Gansu Agricultural University, Yinmen Village, Anning District, Lanzhou 730070, China; 15877595464@163.com (L.L.); lzs0724@163.com (Z.L.); panxj@st.gsau.edu.cn (X.P.); 19119882925@163.com (K.Y.); 18719851354@163.com (Y.W.); 18393427673@163.com (T.Y.); hgh19962024@163.com (G.H.); liaowb@gsau.edu.cn (W.L.)

**Keywords:** multifunctional protein, 3-ketolipoyl-CoA thiolase, fatty acid metabolism, abiotic stresses, tissue expression profiling, gene family, expression analysis

## Abstract

Fatty acids and their derivatives play a variety of roles in living organisms. Fatty acids not only store energy but also comprise membrane lipids and act as signaling molecules. There are three main proteins involved in the fatty acid β-oxidation pathway in plant peroxisomes, including acyl-CoA oxidase (*ACX*), multifunctional protein (*MFP*), and 3-ketolipoyl-CoA thiolase (*KAT*). However, genome-scale analysis of *KAT* and *MFP* has not been systemically investigated in tomatoes. Here, we conducted a bioinformatics analysis of *KAT* and *MFP* genes in tomatoes. Their physicochemical properties, protein secondary structure, subcellular localization, gene structure, phylogeny, and collinearity were also analyzed. In addition, a conserved motif analysis, an evolutionary pressure selection analysis, a *cis*-acting element analysis, tissue expression profiling, and a qRT-PCR analysis were conducted within tomato KAT and MFP family members. There are five *KAT* and four *MFP* family members in tomatoes, which are randomly distributed on four chromosomes. By analyzing the conserved motifs of tomato *KAT* and *MFP* family members, we found that both *KAT* and *MFP* members are highly conserved. In addition, the results of the evolutionary pressure selection analysis indicate that the *KAT* and *MFP* family members have evolved mainly from purifying selection, which makes them more structurally stable. The results of the *cis*-acting element analysis show that *SlKAT* and *SlMFP* with respect may respond to light, hormones, and adversity stresses. The tissue expression analysis showed that *KAT* and *MFP* family members have important roles in regulating the development of floral organs as well as fruit ripening. The qRT-PCR analysis revealed that the expressions of *SlKAT* and *SlMFP* genes can be regulated by ABA, MeJA, darkness, NaCl, PEG, UV, cold, heat, and H_2_O_2_ treatments. These results provide a basis for the involvement of the *SlKAT* and *SlMFP* genes in tomato floral organ development and abiotic stress response, which lay a foundation for future functional study of *SlKAT* and *SlMFP* in tomatoes.

## 1. Introduction

Peroxisomes are multifunctional organelles essential for eukaryotic development and growth and are highly dynamic in both morphology and metabolism [1,2]. The plant peroxisome has a variety of physiological roles, including phytohormone biosynthesis, lipolytic metabolism, and reactive oxygen species metabolism, which are essential for plant development [1,3,4,5]. Fatty acid, which is composed of carbon, hydrogen, and oxygen elements, is ubiquitous in plants, animals, and microorganisms. The catabolism of fatty acids takes place in organisms mainly through the β-oxidation pathway, which, in plants and fungal species, occurs exclusively in peroxisomes [1,6,7,8]. β-oxidation sustains plant growth by degrading fatty acids to acetyl-CoA, which is further metabolized through the glyoxylate cycle [9]. The energy released during these processes can support plant growth activities before the cotyledons turn green to conduct photosynthesis [9]. In addition, fatty acid β-oxidation affects jasmonic acid (JA) synthesis from the precursor linolenic acid [10]. Furthermore, fatty acid β-oxidation also has important roles in the remobilization of reserved energy in the survival of senescence and extended periods of darkness.

The central steps of the fatty acid β-oxidation pathway in the peroxisomes of plants are accomplished by acyl coenzyme oxidase (ACX), multifunctional protein (MFP), and 3-ketoacyl coenzyme a thiolase (KAT) [10]. ACX family members may specifically recognize distinct chain lengths of fatty acids and catalyze the first step of peroxisomal fatty acid β-oxidation to converse fatty acyl-CoAs to trans-2-enoyl CoAs [7,11]. This step is thought to be critical in controlling the rate of carbon flux in the β-oxidation pathway [12]. In addition, different ACX isozymes can target distinct lengths of substrates to promote the efficient utilization of various fatty acids, including long-chain (C14:0 to C20:0), medium-chain (C8:0 to C14:0), short-chain (C4:0 to C8:0), and very-long-chain (C22:0 to C26:0) fatty acids [13]. This diversity plays a role in plant adaptation to different environmental conditions and response to changes in lipid supply. MFP proteins conduct four kinds of enzyme activities, such as 2-transenoyl-CoA hydratase, L-3-hydroxyacyl-CoA dehydrogenase, D-3-hydroxyacyl-CoA epimerase, and Δ3, Δ2-enoyl-CoA isomerase [12]. Moreover, the activities of 2-trans-enoyl-CoA hydratase and L-3-hydroxyacetyl-CoA dehydrogenase are core steps required for the catabolism of all fatty acids [14,15]. KAT members catalyze the final step of fatty acid β-oxidation. As a kind of thiolase, KAT cleaves acetyl-CoA from the fatty acyl chain, resulting in the formation of a new fatty acyl-CoA two carbons shorter than the original substrate [16].

In plants, many MFP and KAT family members have been found. In *A. thaliana*, there are two *MFP* genes involved in β-oxidation, namely, *AIM1* and *AtMFP2* [15]. The corresponding MFP proteins exhibit both 2-trans-enoyl-coenzyme hydratase and L-3-hydroxyacyl-coenzyme dehydrogenase activities [14]. Among them, *AIM1* favors short-chain substrates, while the hydratase activity of *AtMFP2* favors longer-chain fatty acids. Previous studies also found an *MFP* gene in cucumber, and the function of the corresponding protein is unknown [14]. In fungi, there also exist MFP proteins. As a rice blast fungus, *Magnaporthe grisea* MFP1 (MG06148.4) protein is located in peroxisome-like bodies. *M. grisea* MFP1 is required for lipid utilization and fungal virulence. Mutants lacking *MFP1* are unable to use lipids or fatty acids as their sole carbon source, resulting in reduced virulence in plants [17]. In addition, MFP1 is also required for the formation of fully functional adherents, which further affects the appressorium function of the rice blast fungus during the infection of plants. As for *KAT*, there are three genes encoding *KAT* in *A. thaliana*, including *AtKAT1*, *AtKAT2*, and *AtKAT5* [18]. *AtKAT2* is located on chromosome 2 of *A. thaliana*, which encodes a peroxisomal thiolase and mediates seedling morphogenesis during early seedling growth [12,19]. Additionally, *AtKAT2* is an important component of timely senescence in leaves and is responsible for JA biosynthesis in *A. thaliana* [6,20]. Furthermore, AtKAT2-derived β-oxidase influences ABA singalongs and ABA-induced ROS generation [1]. AtKAT5 takes part in producing peroxisomal proteins and cell membrane proteins by alternating transcriptional levels of the related genes [18]. Moreover, AtKAT5 plays a vital role in flower and seed predevelopment. *AtKAT1* is produced through the replication of *AtKAT2* [18], and the function of AtKAT1 as the fatty acid β-oxidation regulator is largely unknown. However, there exist rare studies concerning the members of MFP and KAT in other plants.

The tomato is an important staple and economic crop worldwide, which is also a vital experimental model plant. KAT- and MFP-derived fatty acid β-oxidation has an important role in plant biosynthesis, but their family members have not been well studied in tomatoes. To better understand the role of the *KAT* and *MFP* family genes in tomatoes, we performed secondary structure analysis, conserved motifs, *cis*-acting elements, specific expression in different tissues, evolutionary pressure selection, and covariance analysis. Meanwhile, the expression patterns of *KAT* and *MFP* family members in tomatoes under different treatments of plant hormones and abiotic stresses were investigated. Our study will extend the molecular biological functions of the *KAT* and *MFP* families in plants and suggest potential roles for KAT and MFP family members’ abiotic stress and phytohormone responses.

## 2. Results

### 2.1. Genome-Wide Identification of SlKAT and SlMFP Genes in Tomatoes

In our study, five tomato *KAT* genes and four tomato *MFP* genes were obtained after homologous matching, which were named *SlKAT1*-*SlKAT5* and *SlMFP1*-*SlMFP4* based on the location of the genes on different chromosomes (Table 1).

Amino acid lengths of tomato KAT proteins range between 39,128.82 (SlKAT2) and 50,588.34 (SlKAT4) Da. The amino acid size of the tomato MFP transcription factor family is between 606 aa (SlMFP1) and 723 aa (SlMFP2). The molecular weight is between 67,336.01 (SlMFP1) and 77,793.88 (SlMFP4) Da. By predicting the physicochemical properties, we found that SlKAT1, SlKAT4, and SlKAT5 are basic proteins (PI > 7), and SlKAT2 and SlKAT3 are acidic proteins (PI < 7). In addition, only the GRAVY index of SlKAT5 is below 0 (−0.023). Thus, SlKAT5 is a hydrophilic protein, and the other KAT proteins are hydrophobic. In the MFP family, all the members are basic proteins except for SlMFP3. Moreover, all the MFP family proteins are hydrophobic proteins (Table 1).

### 2.2. Analysis of Conserved Motifs and Gene Structure of the SlKAT and SlMFP Family Genes in Tomatoes

It can be seen from Figure 1 that different members of the *KAT* and *MFP* family genes have different numbers of exons and introns. In the family of *KAT* genes, there are 12 exons and 14 introns in *SlKAT1* and 13 exons and 15 introns in *SlKAT2*. As for *SlKAT3*, *SlKAT4*, and *SlKAT5*, there exist 14 exons and 13 introns. In the family of *MFP* genes, there are 13 exons and 16 introns in *SlMFP1*, 17 exons and 18 introns in *SlMFP2*, 17 exons and 17 introns in *SlMFP3*, and 18 exons and 17 introns in *SlMFP4* (Figure 1).

As a result of the analysis in Figure 2, 10 conserved motifs were identified in tomato KAT proteins. The length of the motifs is between 15 and 50 amino acids (Table 2). All the 10 conserved motifs are presented in SlKAT1, SlKAT2, and SlKAT3. However, motif6 and motif10 are absent in SlKAT4 and SlKAT5. We identified 10 conserved motifs in tomato MFP proteins. The length of the motifs is between 38 and 50 amino acids (Table 3). In contrast to KAT proteins, all the tomato MFP members identically contain 10 conserved motifs.

### 2.3. Phylogenetic Analysis of KAT and MFP Family Members

In order to further understand the phylogenetic relationships between the KAT and MFP family genes in tomatoes, we compared the KAT and MFP protein sequences of tomato, *A. thaliana*, cucumber, and *P. tomentosa* to jointly construct a phylogenetic tree, respectively (Figure 3). The protein sequences of KAT included three from *A. thaliana*, 12 from cucumber, 10 from *P. tomentosa*, and five from tomato. We categorized 30 KAT proteins into four groups (Figure 3). Group A has 6 members, Group B has 7 members, Group C has 4 members, and Group D has 13 members. Among them, SlKAT1 belongs to Group B, SlKAT2 and SlKAT3 belong to Group A, and SlKAT4 and SlKAT5 belong to Group D. Additionally, SlKAT2 has the closest evolutionary relationship with SlKAT3. The protein sequences of MFP included two from *A. thaliana*, seven from cucumber, 21 from *P. tomentosa*, and four from tomato (Figure 3). We found 33 MFP proteins and categorized these proteins into four groups. Group A has nine members, Group B has seven members, Group C has nine members, and Group D has eight members. The protein SlMFP1 belongs to Group A, and SlMFP2, SlMFP3, and SlMFP4 belong to Group B.

### 2.4. Protein Secondary Structure Analysis and Subcellular Localization of Tomato KAT and MFP Family Members

We analyzed the secondary structure of tomato KAT and MFP members (Table 4). The secondary structures in tomato KAT members were composed of the alpha helix (39.71–49.61%), beta turn (7.53–9.45%), and random coil (26.77–36.17%). In addition, the MFP family mainly included the alpha helix, random coil, and beta turn with percentages of 49.37~52.31%, 6.99~7.75%, and 31.67~33.85%, respectively.

The predicted subcellular localization of the tomato showed that the *KAT* and *MFP* genes were mainly distributed in the nucleus, chloroplasts, cytosol, peroxisomes, plasma membrane, and endoplasmic reticulum (Table 5). However, the distribution of each member in different cellular parts was different. *SlKAT5*, *SlKAT1*, *SlMFP3*, and *SlMFP2* were expressed in the nucleus. These genes may be associated with the replication of genetic material. All genes except *SlMFP2* were localized in chloroplasts. Genes localized in chloroplasts may be involved in photosynthesis in tomatoes. In the cytosol, only *SlKAT5* and *SlMFP4* were not involved in expression. *SlKAT1* was not expressed in the peroxisome. Genes other than *SlKAT1* may be involved in some physiological activities of the plant peroxisome. *SlKAT5* and *SlMFP4* were localized in the plasma membrane. In addition, *SlMFP3* and *SlMFP4* were expressed in the endoplasmic reticulum.

### 2.5. Chromosomal Localization of SlKAT and SlMFP Genes in Tomatoes

Using tomato genome annotation information and TBtools software (v2.034), we visualized the chromosomal distribution of tomato *SlKAT* and *SlMFP* family members. We found that the genes for tomato *SlKAT* and *SlMFP* are unevenly distributed on the chromosomes (Figure 4). Among them, *SlKAT1* is located on chromosome 4, *SlKAT2* is located on chromosome 5, *SlKAT3* is located on chromosome 7, and *SlKAT4* and *SlKAT5* are located on chromosome 9. *SlMFP1* is located on chromosome 1, *SlMFP4* is located on chromosome 7, *SlMFP3* is located on chromosome 8, and *SlMFP4* is located on chromosome 12.

### 2.6. Analysis of Cis-Acting Elements of the Tomato SlKAT and SlMFP Family Genes

We analyzed promoter sequences (from −2000 bp to −1 bp) of five *SlKAT* genes and four *SlMFP* genes to detect the *cis*-acting elements (Figure 5). There were 20 major *cis*-acting elements in the promoter regions of tomato *KAT* family members. The light-responsive elements included I-box, MRE, G-box, G-Box, Box 4, GATA-motif, TCT-motif, and ACE. The hormone response elements included GARE-motif, CGTCA-motif, TGACG-motif, ABRE, TATC-box, P-box, and TCA-element. The stress response components included circadian, MBS, ARE, TC-rich repeats, and LTR. Within the *MFP* family, we identified 16 major *cis*-acting elements. Among them, MRE, Box 4, circadian, GT1-motif, G-Box, G-box, Sp1, and GA-motif belonged to the light-responsive elements. ABRE, GCN4_motif, TGA-element, TGACG-motif, CGTCA-motif, and GARE-motif belonged to hormone-responsive elements. ARE and TC-rich repeats belonged to the stress-responsive elements.

As can be seen in Figure 6, in the *SlKAT* family, Box 4 elements are present in all members. *SlKAT4* responded most significantly to the Box 4 element. *SlKAT5* responded most significantly to the CGTCA-motif and TGACG-motif elements. *SlKAT2* responded most significantly to the G-box and ABRE elements. *SlKAT1* responded most significantly to the CGTCA-motif and TGACG-motif elements. In addition, in the *SlMFP* family, Box 4 elements are present in all members. *SlMFP1* and *SlMFP3* primarily respond to Box 4 elements. *SlMFP4* responds more significantly to G-box, ABRE, TGACG-motif, and CGTCA-motif elements. *SlMFP2* primarily responds to Box 4 elements. The results of the *cis*-acting elements indicate that tomato *SlKAT* and *SlMFP* family members function in light-responsive elements, hormone-responsive elements, and stress-responsive elements.

### 2.7. Tissue Expression Analysis of Tomato SlKAT and SlMFP Family Genes

In order to investigate the role of the *SlKAT* and *SlMFP* genes in various tomato tissues during various growth stages, the expression levels of the *SlKAT* and *SlMFP* genes in 14 tomato tissues were analyzed, including Unopened Flower Bud, Fully Opened Flower, Leaf, Root, 1 cm Fruit, 2 cm Fruit, 3 cm Fruit, Mature Green Fruit, Breaker Fruit, Breaker Fruit + 10, Pimpinellifolium Immature Green Fruit, Pimpinellifolium Breaker Fruit, Pimpinellifolium Breaker + 5 Fruit, and Pimpinellifolium Leaf (Figure 7). Among them, *SlKAT5* and *SlMFP4* were highly expressed in all the detected tissues. In addition, the transcript abundance of *SlKAT5* was most abundant in Pimpinellifolium Breaker + 5 Fruit. *SlMFP4* was highly expressed in Unopened Flower Bud and Fully Opened Flowers. *SlKAT4* expression was higher in Unopened Flower Bud, whereas *SlKAT2* and *SlMFP3* expression was higher in 1 cm Fruit and 2 cm Fruit. The expression of *SlKAT3*, *SlKAT1*, and *SlMFP1* was low in the other tissues. *SlMFP2* expression was higher in Mature Green Fruit, Breaker Fruit, Breaker Fruit + 10, Pimpinellifolium Breaker Fruit, and Pimpinellifolium Breaker + 5 Fruit.

### 2.8. Gene Duplication and Collinearity Analysis of the SlKAT and SlMFP Genes

In order to understand the amplification mechanism of the *KAT* and *MFP* family genes, we investigated the covariance of *KAT* and *MFP* in the tomato, *A. thaliana*, and *P. tomentosa* (Figure 8). We found that tandem duplicate events occurred for *SlKAT4* and *SlKAT5*. This can impact the level and pattern of gene expression as well as lead to genetic instability. To investigate the evolutionary relationship between the *KAT* and *MFP* family genes among species, we analyzed the homology of the *KAT* and *MFP* genes in the tomato, *A. thaliana*, and *P. tomentosa*. Among the *KAT* gene family, four genes were linearly related between tomato and *P. tomentosa*. One gene in *A. thaliana* was covariantly related to tomato. Three genes were linearly related between *P. tomentosa* and tomato. Four genes were linearly related between *A. thaliana* and tomato. The *SlKAT5* gene was co-localized in both *P. tomentosa* and *A. thaliana*. This indicates that the *SlKAT5* gene was present before the differentiation of tomato, *A. thaliana*, and *P. tomentosa* (Figure 8A).

### 2.9. Evolutionary Selection Pressure Analysis of SlKAT and SlMFP Genes

Ka denotes the non-synonymous substitution rate, which is the incidence of non-synonymous mutations in the gene sequence. Ks denotes the synonymous substitution rate, which is the incidence of synonymous mutations in a gene sequence [21]. The Ka/Ks values can be used to measure the relative frequency of nonsynonymous and synonymous mutations, thus reflecting the selective pressure on genes during evolution [22]. It has an important role in the evolutionary analysis of family genes. Here, we performed Ka/Ks analysis on the *SlKAT* and *SlMFP* family genes, and we found that the Ka/Ks values in both *SlKAT* and *SlMFP* genes were less than 1 (Figure 9). This indicates that the selective pressure for the evolution of the *SlKAT* and *SlMFP* family genes in tomato species stems mainly from purifying selection.

### 2.10. Expression Profile Analysis of the SlKAT and SlMFP Genes in Response to Hormone and Abiotic Stresses

To explore the response patterns in the *SlKAT* and *SlMFP* genes to different hormones, we investigated the expression levels of five *SlKAT* genes and four *SlMFP* genes in the tomato under ABA and MeJA. In *SlKAT* family members, the expression levels of *SlKAT1* and *SlKAT2* were significantly suppressed by the ABA treatment. However, the expression levels of *SlKAT3*, *SlKAT4*, and *SlKAT5* were increased by the 6–24 h ABA treatment. Under the MeJA treatment, the expression levels of *SlKAT1* and *SlKAT4* were significantly down-regulated. Also, the 24 h MeJA treatment restrained the expression abundance of *SlKAT2*. However, the expression levels of *SlKAT3* and *SlKAT5* were enhanced to 2.04-fold and 2.98-fold, respectively, under the 12 h MeJA treatment. As for the *SlMFP* family genes, the expression of *SlMFP1*, *SlMFP2*, and *SlMFP3* were increased by the 12 h ABA treatment (Figure 10). In addition, the 6 and 24 h MeJA treatments significantly repressed the transcriptional levels of *SlMFP* genes.

The expressions of the *SlKAT* and *SlMFP* genes were also detected under darkness, NaCl, PEG, UV, cold, heat, and H_2_O_2_ treatments (Figure 11). In the *SlKAT* family, the expression levels of *SlKAT1* and *SlKAT4* were down-regulated after dark treatment. However, the expression levels of *SlKAT3* and *SlKAT5* reached a peak at 12 h of dark treatment. The expression levels of *SlKAT1*, *SlKAT3*, *SlKAT4*, and *SlKAT5* were up-regulated by NaCl treatment, in which expressions of *SlKAT1* and *SlKAT3* were highest at 12 h of NaCl treatment. However, the transcriptional levels of *SlKAT2* were down-regulated by NaCl treatment. Similarly, the expression levels of *SlKAT1*, *SlKAT3*, *SlKAT4*, and *SlKAT5* were up-regulated under PEG conditions. UV treatment resulted in up-regulation of the expression levels of *SlKAT3*, *SlKAT4*, and *SlKAT5* at 6 h and 12 h. The expression levels of *SlKAT1* were up-regulated at 12 h and 24 h of UV treatment, and *SlKAT2* peaked at 12 h of UV treatment. Under cold treatment, the expression levels of *SlKAT1* and *SlKAT2* were down-regulated, whereas the transcriptional levels of *SlKAT4* were up-regulated. The highest expression levels of *SlKAT3* and *SlKAT5* were observed after 6 h of cold treatment. Heat treatment caused the expression of *SlKAT3*, *SlKAT4,* and *SlKAT5* to be up-regulated. The expression levels of *SlKAT1* and *SlKAT2* were highest at 6 h of heat treatment. The result of the H_2_O_2_ treatment led to a down-regulation of *SlKAT2* expression. The expression levels of *SlKAT1*, *SlKAT3*, *SlKAT4*, and *SlKAT5* peaked as a result of 6 h of H_2_O_2_ treatment. In the *SlMFP* family, the transcriptional levels of all the *SlMFPs* were down-regulated after dark treatment. After treatment with NaCl, the expression levels of *SlMFP1* were down-regulated, and *SlMFP2* had the lowest expression at 6 h. The expression levels of *SlMFP2* and *SlMFP4* were highest at 12 h under PEG treatment. UV treatment up-regulated the transcript abundances of all *SlMFPs* genes. The expression levels of *SlMFP1*, *SlMFP2*, and *SlMFP4* were down-regulated under cold treatment. Under heat treatment, *SlMFP1* and *SlMFP3* had the highest expression at 6 h. Under H_2_O_2_ conditions, the transcriptional level of *SlMFP1* was highest at 12 h, whereas the expression levels of *SlMFP2*, *SlMFP3*, and *SlMFP4* were highest at 6 h.

## 3. Discussion

There exist three key proteins involved in the fatty acid β-oxidation pathway in plant peroxisomes, including ACX, MFP, and KAT. Together, these enzymes catalyze the complete degradation of both saturated and unsaturated long-chain fatty acyl-CoAs to acetyl-CoA via the repeated cleavage of acetate units from the thiol end of fatty acids [23,24]. In previous studies, *KAT* and *MFP* genes were synergistically expressed during seed germination in *A. thaliana* and shared a common regulatory mechanism [10]. Alternatively, KAT and MFP can affect photosynthetic capacity by influencing sucrose conversion during photosynthesis [10]. The present study was conducted to analyze the tomato *SlKAT* and *SlMFP* family genes bioinformatically, and a total of five *SlKAT* genes and four *SlMFP* genes were identified (Table 1). Nevertheless, there are only three *KAT* genes and two *MFP* genes in *A. thaliana* [15,18]. Genome-wide replication multiplies the number of genes, which in turn leads to a multiplication of gene family members [25,26,27]. Hence, the enhanced numbers of *SlKAT* and *SlMFP* families may be caused by genome-wide replication. Moreover, the *KAT* and *MFP* genes were randomly distributed in tomato and *A. thaliana* chromosomes (Figure 4). As the differences in genome size across different species may lead to variations in the number of gene families, the enhanced members of the *SlKAT* and *SlMFP* families may be also caused by the larger genome size of the tomato than *A. thaliana* [25]. Meanwhile, *SlKAT4* and *SlKAT5* are located on the same chromosome and have the same number of motifs with similar distributions (Figure 2 and Figure 4). In addition, the results of the analysis of collinearity indicated a linear relationship between *SlKAT4* and *SlKAT5* (Figure 8A). Therefore, the increase in *SlKAT* genes may be related to gene duplication, which is similar to a previous study [25]. The collinearity analysis of the genes revealed that *SlKAT5* is homologous in both *A. thaliana* and *Populus tomentosa*, which indicates that the *SlKAT5* gene may have been inherited from an earlier plant, similar to a previous study (Figure 8B). Meanwhile, we analyzed the phylogenetic relationships within *SlKAT* and *SlMFP* members, respectively. The universal phylogenetic tree not only spans all extant life, but its root and earliest branching represent stages in the evolutionary process before modern cell types came into being [28]. We found that two *SlKAT* proteins (*SlKAT2* and *SlKAT3*) and two *SlMFP* proteins (*SlMFP2* and *SlMFP4*), respectively, clustered on the same branch of the phylogenetic tree (Figure 3), suggesting that these proteins are closely related.

In *A. thaliana*, AtMFP2 is a 79-kDa protein composed of three domains and a linker helix [14]. It forms a stable monomeric structure and resembles the mammalian peroxisomal MFE-1 in domain organization and size [14]. Nevertheless, the secondary structures of the SlKAT and SlMFP families in this study are predominantly α-helices (Table 4). In proteins, the α-helix plays a role in supporting and stabilizing the molecular structure [29,30]. Thus, the α-helices may lead to greater structural stability within the *SlKAT* and *SlMFP* family members. Genetic selection pressures contribute to understanding evolutionary relationships. Ka denotes the non-synonymous substitution rate, which is the incidence of non-synonymous mutations in the gene sequence [21]. Ks denotes the synonymous substitution rate, which is the incidence of synonymous mutations in a gene sequence [21]. Ka/Ks values can be used to measure the relative frequency of nonsynonymous and synonymous mutations, thus reflecting the selective pressure on genes during evolution [22]. Ka/Ks values less than 1 represent genes mainly from purification selection. We performed Ka/Ks analysis on the *SlKAT* and *SlMFP* family genes, and we found that the Ka/Ks values in both the *SlKAT* and *SlMFP* genes were less than 1 (Figure 9). This indicates that the *SlKAT* and *SlMFP* family genes in tomato species stem mainly from purifying selection, which makes them more evolutionarily conserved, structurally stable, and potentially more functionally consistent. Thus, *SlKAT* and *SlMFP* members are more evolutionarily conserved and structurally stable, suggesting that they may be functionally aligned. In the analysis of conserved motifs in tomato SlKAT and SlMFP proteins, we found that the 10 conserved motifs identified were neatly aligned from n-terminal to c-terminal and fixed in position (Figure 2). Hence, SlKAT and SlMFP members may be highly conserved. As for gene structure, the *SlKAT* genes and the *SlMFP* genes have a similar exon–intron structure within their family members in tomatoes (Figure 1). In previous studies, the gene structures of *MFP* family members were found to be highly conserved in *A. thaliana* [15]. Moreover, *AIM1* and *AtMFP2* have both 2-trans-enoyl-coenzyme hydrogenase and L-3-hydroxyacetyl-coenzyme dehydrogenase activities in *A. thaliana*. In cucumber, the *CsMFPa* gene shares 56% sequence identity with *AtMFP2*, as well as 75% sequence identity with *AIM1* over its gene sequence [14]. Similarly, CsMFPa possesses similar enzyme activities as AIM1 and AtMFP2 [14]. Hence, the similar gene sequences of the *SlKAT* and *SlMFP* genes suggest a similar protein function in SlKAT and SlMFP members, respectively, which need further identification.

According to the results of tissue-specific expression, the *SlKAT* and *SlMFP* family genes were expressed in all the detected tomato tissues, including the leaves, roots, flowers, and fruits of different developmental stages, except for *SlMFP1* (Figure 7). There are three genes encoding *KAT* in *A. thaliana*, including *AtKAT1*, *AtKAT2*, and *AtKAT5*. In particular, *AtKAT2* is highly expressed throughout the life cycle, especially in the later stages of flower and seed development [18]. Among the *SlKAT* family members, *SlKAT5* was highly expressed in various fruit development stages, including Breaker Fruit, Breaker Fruit + 10, Pimpinellifolium Breaker Fruit and Pimpinellifolium Breaker + 5 Fruit (Figure 7). ABA has an important effect on the accumulation of flavonoids and carotenoids during tomato fruit ripening [31]. In addition, the presence of ABA increases fruit firmness by modifying the tomato cell wall [32], and *AtKAT2* has a positive regulatory effect on ABA signaling [33]. In our study, *SlKAT5* was significantly up-regulated in response to ABA treatment. Thus, *SlKAT5* may take part in plant reproductive development, which may imply the role of *SlKAT5* in tomato yield regulation. However, whether *SlKAT5* regulates fruit ripening by directly participating in the ABA response needs to be uncovered in the future. In addition, the expression level of *A. thaliana AtKAT5* is high in young flowers [18]. In our study, the transcriptional level of *SlKAT4* was the most abundant in unopened flower buds (Figure 7). Thus, the physiological functions of tomato *SlKAT4* and *A. thaliana AtKAT5* in the development of floral organs may be identical, which needs to be studied further. In *A. thaliana*, the functional deletion of AIM1 resulted in abnormal inflorescence development and abnormal formation of inflorescence meristem organization [15]. In our study, the expression level of *SlMFP4* was the most abundant in Unopened Flower Buds and Fully Opened Flowers (Figure 7), suggesting a similar role of SlMFP4 and AIM1 in plant floral organ development. In addition, whether the absence of the *SlMFP4* gene will lead to abnormal development of tomato floral organs, thereby affecting plant fertility, requires further in-depth research. It should be noted that the flowering and fruiting of plants are complex processes regulated by various factors such as light, temperature, water, and nutrients. In practical applications, it is necessary to consider the plant’s growth environment and physiological status, as well as the complexity of gene regulation, in order to better achieve the goal of increasing yield. However, the accurate activities of *SlKAT* and *SlMFP* family members within leaves, roots, flowers, and fruits during tomato growth and development still need to be investigated.

The β-oxidation enzyme KAT2 has been reported to promote ABA responses, including inhibition of seed germination, post-germinative growth arrest, and stomatal closure in *A. thaliana* [1]. In plants, β-oxidation is a source of H_2_O_2_ in peroxisomes, and ROS (including H_2_O_2_) are known to be involved in ABA signaling [1]. In addition, ABA negatively regulates KAT2-mediated ROS generation [33]. In the studied tomatoes, the expression levels of *SlKAT3*, *SlKAT4*, and *SlKAT5* were significantly up-regulated under ABA treatment (Figure 10A). Thus, SlKAT may be regulated by ABA during tomato seedling growth. It is known that fatty acid β-oxidation is involved in the synthesis of the lipid signaling molecule MeJA from acrylic acid [16,19,34]. MeJA is a methyl-esterified product of JA, which is mainly involved in regulating insect resistance and responses to adverse stress in plants [6,20,35]. In the present study, the results of the *cis*-acting elements showed that *SlKAT5* and *SlMFP4* responded most significantly to MeJA (Figure 6). However, we found that the expressions of *SlKAT* with *SlMFP* family members were not up-regulated under MeJA treatment, except for *SlKAT3* and *SlKAT5* under 12 h of treatment. (Figure 10B). The results of MeJA treatment were not consistent with those of the *cis*-acting element. Therefore, the role of *SlKAT* and *SlMFP* in the regulation of MeJA in tomatoes remains to be investigated.

By now, the role of SlKAT and SlMFP members in abiotic stresses has not been thoroughly studied in tomatoes. In the present article, the influences of various abiotic stresses on the transcript abundance of *SlKAT* and *SlMFP* genes were detected. Light is crucial for plant life, and the perception of light dictates plant growth, morphology, and developmental changes [36]. In our study, there were eight *cis*-acting components in the *SlKAT* and *SlMFP* families, respectively, that deal with light response. Except for *SlKAT5*, the expression of other genes in tomato *SlKAT* and *SlMFP* was down-regulated under dark conditions (Figure 11A). Hence, we hypothesize that under dark conditions, the expressions of the *SlKAT* and *SlMFP* genes are reduced, which may lead to a decrease in fatty acid β-oxidation activities [37]. Salt stress decreases the photosynthesis and respiration rates of plants [38]. Drought is one of the major constraints limiting crop production worldwide [39]. Enhanced fatty acid β-oxidation induces ROS production, which can negatively affect plant survival in unfavorable environments [37]. Increased fatty acid access to the β-oxidation pathway decreases plant resistance to salt stress. We found that *SlKAT4* and *SlKAT5* were significantly induced by NaCl (Figure 11B) and drought stress (Figure 11C). Thus, we hypothesize that *SlKAT4* and *SlKAT5* play a great role in responding to salt stress and drought stress in tomatoes. There are three proteins (ACX, MFP, and KAT) that are involved in fatty acid β-oxidation, and it is known that UV treatment up-regulates *ACX* gene expression [24]. While *ACX* genes function to alleviate UV stress, *SlKAT* and *SlMFP* family members may have the same response pattern to UV. This conjecture can be verified by Figure 11D, which shows that all gene expressions except *SlKAT2* were up-regulated. Tomato plants have limited ability to acclimatize to temperature changes [40]. Under cold treatment conditions (Figure 11E), the expressions of the *SlKAT1/2* and *SlMFP4* were up-regulated. In contrast, under heat-treated conditions (Figure 11F), the expression of the *SlKAT3/4/5* and *SlMFP3* was up-regulated. As the expression patterns of *SlKAT* and *SlMFP* were opposite under the cold and heat treatments, we conclude that the *KAT* and *MFP* family genes in tomatoes may be involved in physiological activities under low-temperature environments and have some mitigating effect on high-temperature stress. However, the specific functions of *SlKAT* and *SlMFP* under cold- and high-temperature stress need to be investigated further. The fatty acid β-oxidation pathway is the main source of H_2_O_2_ in peroxisomes [2,5,10]. Catalase plays a role in the production of hydrogen peroxide by peroxisomes, and three genes encoding catalases have been identified in *A. thaliana* and other plants, of which *CAT1* is associated with fatty acid β-oxidation [41]. The activity of the fatty acid β-oxidases KAT and MFP may affect the expression of the *CAT1* gene, which in turn affects catalase activity and hence, hydrogen peroxide production. The expression of *SlKAT3*, *SlKAT4*, *SlKAT5*, and *SlMFP1* was up-regulated under H_2_O_2_ treatment (Figure 11G). We hypothesize that these four genes play a role in alleviating H_2_O_2_ stress. In addition, ACX4, KAT2, and MFP2 proteins involved in fatty acid β-oxidation are also involved in the ROS1-mediated DNA demethylation process in *A. thaliana* [42]. What is more, DNA demethylation in plants plays a role in many important processes, including fruit ripening and biotic and abiotic stress responses [43]. In our study, the results of the tomato tissue expression analysis showed that *SlKAT5* genes were highly expressed in fruits (Figure 7), while the expression of *SlKAT4*, *SlKAT5*, and *SlMFP1* was obviously altered under the salt stress treatment (Figure 10). Thus, our study may provide some theoretical basis for *KAT* and *MFP* genes in tomato fruit development and the mitigation of salt stress. However, whether KAT and MFP proteins regulate fruit development and the salt stress response through DNA demethylation still needs to be discovered. It is well known that understanding the intricate relationship between light conditions, gene expression, and physiological responses in plants is essential for optimizing growth conditions and enhancing crop productivity. By investigating these interactions, we can gain valuable insights into the molecular mechanisms underlying plant responses to environmental stimuli, ultimately contributing to strategies for improving biomass and yield in agricultural settings. Thus, the mechanism by which tomato *SlKAT* and *SlMFP* respond to light may be important for fruit yield and needs to be further uncovered. Meanwhile, the tomato *SlKAT* and *SlMFP* gene families have important potential functions in hormonal and abiotic stress responses. This may be a focus for future research, but the specific functions of these genes remain to be investigated by further studies.

## 4. Materials and Methods

### 4.1. Genome-Wide Identification of MFP and KAT Gene Family Members in Tomatoes

Firstly, the tomato genome sequence and ITG4.0 annotation information were downloaded from the online database Phytozome v13 (https://phytozome.jgi.doe.gov/pz/ (accessed on 6 May 2023)), [44]. Then, tomato KAT and MFP protein sequences were extracted by using ‘GXF sequence extract’ and ‘Batch Translate CDS’ in TBtools (Toolbox for ecological battle) v2.034 software. The *A. thaliana* KAT and MFP protein sequences were downloaded from the online database TAIR (https://www.arabidopsis.org/index.jsp (accessed on 6 May 2023)). The collected tomato and *A. thaliana* protein sequences were then subjected to unidirectional BLAST using the ‘BLAST GUI Wrapper’ function in TBtools software (v2.034), and the corresponding bidirectional BLAST was performed using the NCBI database (https://www.ncbi.nlm.nih.gov/ (accessed on 6 May 2023)). Next, the ‘Blast Xml to Table’ function in TBtools software (v2.034) was utilized to obtain the possible gene family members of tomato *SlMFP* and *SlKAT*. In detail, the structural domains of the MFP and KAT proteins of the tomato were verified using the SMART database (http://smart.embl-heidelberg.de/ (accessed on 6 May 2023)), and the tomato *SlMFP* and *SlKAT* gene family members were screened according to the conserved structural domains [45,46].

### 4.2. Gene Structure and Chromosomal Localization

Firstly, we analyzed the gene structures of each member of *SlKAT* and *SlMFP* using the ‘Visualize Gene Structure (from GTF/GFF3 File)’ function in TBtools software (v2.034) [47,48]. Then, the gene location was visualized by using the ‘Gene Location Visualize from GTF/GFF file’ function in Tbtools software (v2.034). Finally, the chromosomal location of each tomato *SlKAT* and *SlMFP* gene member was mapped to the tomato genome and renamed based on the chromosomal distribution.

### 4.3. Gene Structure and Protein Conserved Domain Analysis

The gene structure of each member in the SlKAT and SlMFP families was analyzed using the ‘Visualize Gene Structure (from GTF/GFF3 File)’ function in TBtools software (v2.034). All the protein sequences of tomato SlKAT and SlMFP were entered using the MEME (http://meme-suite.org/tools/meme (accessed on 13 May 2023)) online program and analyzed for conserved motifs of the tomato *SlKAT* and *SlMFP* family genes [49,50]. Finally, the conserved motifs of tomato *SlKAT* and *SlMFP* members were visualized and analyzed by the ‘Gene Structure View (Advances)’ function in TBtools software (v2.034).

### 4.4. Phylogenetic Analysis

The KAT and MFP protein sequences of *A. thaliana*, cucumber, and *Populus tomentosa* were downloaded via the online database TAIR and Phytozome v13 (https://phytozome.jgi.doe.gov/pz/ (accessed on 13 May 2023)). The phylogenetic trees containing 30 KAT protein sequences and 33 MFP protein sequences were, respectively, constructed using the neighbor-joining method in MEGA 7.0 software (v7.0.26) (Bootstrap parameter set to 1000) [51]. Additionally, the EvolView (https://evolgenius.info//evolview-v2/#login (accessed on 13 May 2023)) website was used to beautify the evolutionary trees.

### 4.5. Cis-Acting Element Analysis of the Tomato SlKAT and SlMFP Genes

We extracted and analyzed promoter sequences 2000 bp upstream of the tomato *SlKAT* and *SlMFP* genes. The sequences were analyzed for *cis*-acting elements through the Plant CARE online website (http://bioinformatics.psb.ugent.be/webtools/plantcare/html/(accessed on 18 May 2023)) and visualized using Tbtools [52].

### 4.6. Characterization of SlKAT and SlMFP Genes in Tomatoes

The data on chromosomal location, amino acid length, molecular weight, electric point, molecular formula, and other physicochemical characteristics of the tomato KAT and MFP protein sequences were studied through the online website Expasy (https://web.expasy.org/protparam/ (accessed on 18 May 2023)). The prediction of subcellular locations of the tomato *KAT* and *MFP* genes was analyzed online using WoLFPSORT (https://wolfpsort.hgc.jp/ (accessed on 18 May 2023)). The secondary structures of tomato KAT and MFP family proteins were examined using the online website prabi (http://www.prabi.fr/ (accessed on 18 May 2023)), and the relevant data were exported and imaged [47,48].

### 4.7. Analysis of Tissue Expression of SlKAT and SlMFP Genes in Tomatoes

Tissue expression information for the tomato *SlKAT* and *SlMFP* genes was obtained from the eFP (http://bar.utoronto.ca/efp/cgi-bin/efpWeb.cgi (accessed on 1 July 2023)) database. The eFP (http://bar.utoronto.ca/efp/cgi-bin/efpWeb.cgi (accessed on 1 July 2023)) database was searched for the IDs of the *SlKAT* and *SlMFP* genes. We obtained tissue-specific expression analysis information from the eFP database using the tomato *SlKAT* and *SlMFP* gene IDs. We organized data on the expression of *SlKAT* and *SlMFP* in different tissues of tomato and then used Tbtools to map the expression patterns of *SlKAT* and *SlMFP* in different tissues.

### 4.8. Ka (Nonsynonymous)/Ks (Synonymous) Analysis and Colinearity Analysis

The relationship between Ka (non-synonymous substitution rate) and Ks (synonymous substitution rate) was calculated using the CDS sequences and protein sequences of tomato KAT and MFP by using the ‘Simple Ka/Ks calculator (NG)’ function in TBtools software (v2.034) [47,48]. The online database Phytozome v13 (https://phytozome.jgi.doe.gov/pz/ (accessed on 1 July 2023)) was used to download genome files and annotation files for tomatoes and *A. thaliana*. The covariance between tomato and *A. thaliana* was then analyzed using the ‘One Step MCScanX’ function in Tbtools. Finally, the covariance between tomato and *A. thaliana* was visualized using the Advanced Circus function in Tbtools [22].

### 4.9. Plant Materials, Growth Conditions, and Stress Treatments

‘Micro-Tom’ is a tomato variety widely used in scientific research. ‘Micro-Tom’ is characterized by its small size, short growth cycle, small fruits, and a high degree of self-fertilization [53]. Here, ‘Micro-Tom’ tomato seeds with full grains and consistent sizes were selected and put into a 50 mL centrifuge tube and were surface sterilized with 1% NaClO solution for 10 min. The sterilized seeds were put into a 250 mL conical flask filled with 100 mL sterile water and then placed in a HYG-C type shaker and cultured at a rotation speed of 180 r min^−1^ at 25 °C for 3 days. The sterile water was changed once a day. The germinated tomato seeds were planted in a hole dish containing nutrient soil and placed in a growth chamber. The light intensity in the growth chamber was 250 mol photons m^−2^ s^−1^, and the temperature was 26 ± 2 °C for 16 h during the day and 20 ± 2 °C for 8 h at night. The relative humidity was 60%. Plant material was grown for a fortnight and then transplanted from soil to nutrient solution for hydroponics. Then, 21-day-old seedlings with uniform size were used for the subsequent treatments.

For the salt, ABA, H_2_O_2_, MeJA, and drought stress treatments, the selected seedlings were transferred to 1/2 nutrient solution containing 200 mM NaCl, 100 mM ABA, 10% (*w*/*v*) (2.94 M) hydrogen peroxide (H_2_O_2_), 100 mM MeJA, and 20% (*w*/*v*) PEG6000, respectively. All the seedlings were grown in the same conditions as the growth chamber. For the cold and heat treatments, the seedlings were grown in the same 1/2 nutrient solution and were put into growth chambers at 4 °C and 40 °C, respectively. For the dark treatment, the seedlings were moved into a dark growth chamber with the same growth condition as above. For the UV treatment, the selected seedlings were transferred to a growth chamber equipped with 253.7 nm UV-C radiation by a UV-C lamp (Philips Corporation, Poland), and the other growth conditions were the same as the control. After 6, 12, and 24 h, the aboveground parts of the treated seedlings in each replication were harvested separately, frozen with liquid nitrogen, and then stored at −80 °C, respectively. The plants in the 0 h treatment of each treated group were used as controls. The materials for the control group and the experimental group were stored in the same manner. Each treatment of each treated time contained three biological replicates, and each replicate consisted of eight seedlings. All the plants were tested at the same seedling stage to ensure that treatment effects were assessed at the same physiological stage. The experimental groups varied in their different treatments and treatment times.

### 4.10. RNA Isolation and qRT-PCR

Total RNA was extracted from the samples using TRIzol reagent (Invitrogen, Carlsbad, CA, USA) [26,27]. The purity and concentration of RNA were then examined by a Pultton P100_+_ ultra-micro spectrophotometer (Wuzhou Dongfang, Beijing, China). The A260/A280 ratios of the RNA samples between 2.0 and 2.1 were chosen for the subsequent experiments. Then, the FastQuant First Strand cDNA Synthesis Kit (Tianen, Beijing, China) was used to synthesize cDNA. These reactions were executed under the following conditions: 37 °C for 15 min, 85 °C for 5 s, and then finally ended at 4 °C. The SYBR Green Premix Pro Taq HS Premix kit was used for qRT-PCR with a LightCycler 480 Real-Time PCR System (Roche Applied Science, Penzberg, Germany). The qRT-PCR reaction system contained 10 μL 2 × SYBR Green Pro Taq HS Premix, 0.4 μL primer F, 0.4 μL primer R, 2 μL cDNA, and 7.2 μL ddH_2_O. The primers used in qRT-PCR were designed with Primer Premier 5.0 (Premier Biosoft Corporation, USA), and the internal reference was *SlActin* (NC015447.3), which showed a stable expression level among all the samples tested in our previous study [45]. The sequences of the above primers are listed in Appendix A. The 2^−∆∆CT^ calculation method was used to quantify the relative expression of each gene as described in Schnittger, T.D. [54]. The relative expression values of each gene under each treatment at different treated times were calculated by comparing them with those at 0 h. All experiments in our study were repeated three times independently to ensure the reliability and statistical significance of the results.

## 5. Conclusions

In this study, we identified a total of five *KAT* and four *MFP* genes in tomatoes and analyzed their physicochemical properties, secondary structure, conserved motifs, *cis*-acting elements, evolutionary relationships, covariance, evolutionary selection pressures, and expression patterns. We found that tomato *KAT* and *MFP* genes are highly conserved and have important roles in mitigating adversity stresses and hormone responses. Our research proposes the roles of *KAT* and *MFP* genes in tomato growth and adversity stress responses, providing a theoretical foundation for further exploration of fatty acid β-oxidases in tomato plants.

## Figures and Tables

**Figure 1 ijms-25-02273-f001:**
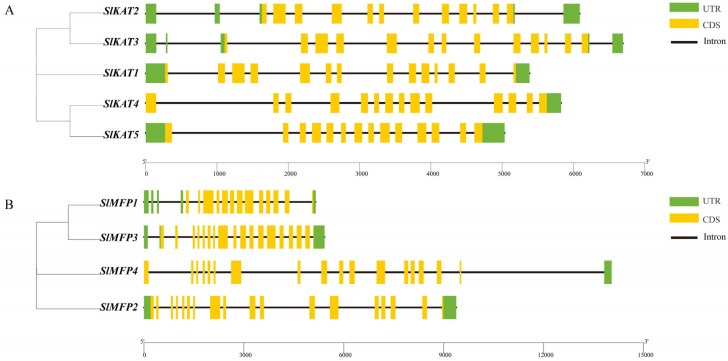
Exon–intron structure of tomato *SlKAT* and *SlMFP* genes. (**A**) The exon–intron mapping of tomato *KAT* genes using TBtools. Green rectangles represent exons and orange rectangles represent upstream and downstream noncoding regions of genes. Black solid lines represent introns. (**B**) The exon–intron mapping of tomato *MFP* genes using TBtools. Green rectangles represent exons and orange rectangles represent upstream and downstream noncoding regions of genes. Solid black lines represent introns.

**Figure 2 ijms-25-02273-f002:**
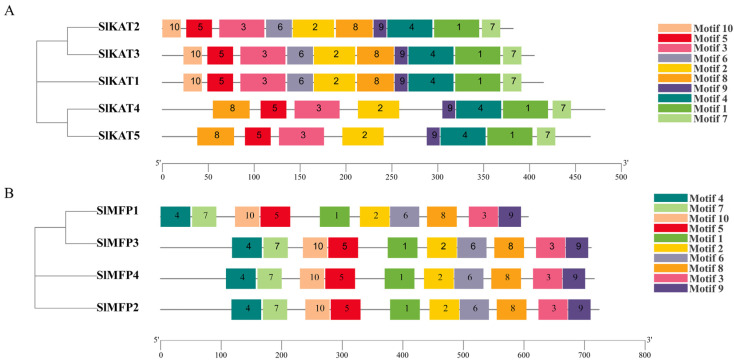
Composition of conserved motifs in tomato KAT and MFP proteins. (**A**) Sequence analysis of the tomato KAT family. Different colors represent different conserved motifs. (**B**) Sequence analysis of the tomato MFP family. Different colors represent different conserved motifs.

**Figure 3 ijms-25-02273-f003:**
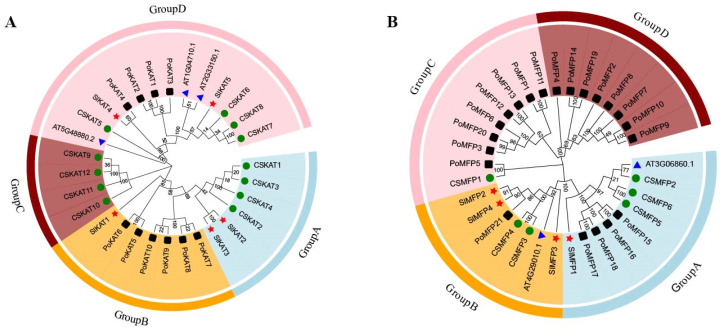
Phylogenetic tree of the KAT and MFP family genes in tomato, *A. thaliana*, *P. tomentosa*, and cucumber. (**A**) Phylogenetic tree of KAT members. KAT protein sequences from *A. thaliana* (3 proteins), cucumber (12 proteins), *P. tomentosa* (10 proteins), and tomato (5 proteins). The tomato is marked with a red pentagram. (**B**) Phylogenetic tree of MFP members. MFP protein sequences from *A. thaliana* (2 proteins), cucumber (7 proteins), *P. tomentosa* (21 proteins), and tomato (4 proteins). The tomato is marked with a red pentagram. The *P. tomentosa* is marked with a black square. The *A. thaliana* is marked with a blue triangle. The cucumber is marked with a green circle.

**Figure 4 ijms-25-02273-f004:**
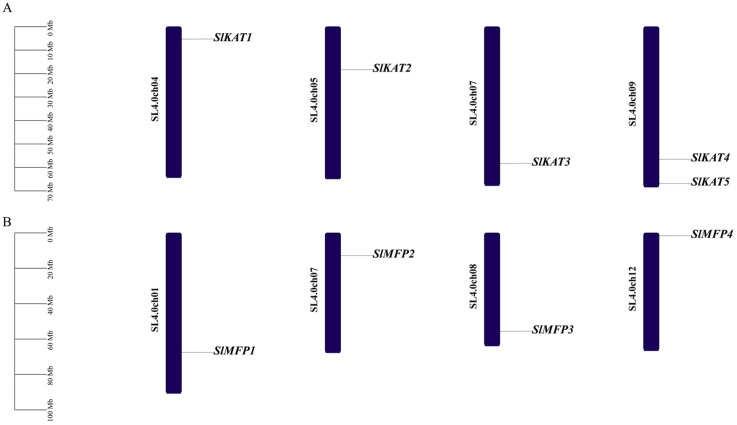
Chromosomal distribution of members of the *SlKAT* and *SlMFP* family genes in the tomato. (**A**) Chromosomal localization of *KAT* genes. (**B**) Chromosomal localization of *MFP* genes.

**Figure 5 ijms-25-02273-f005:**
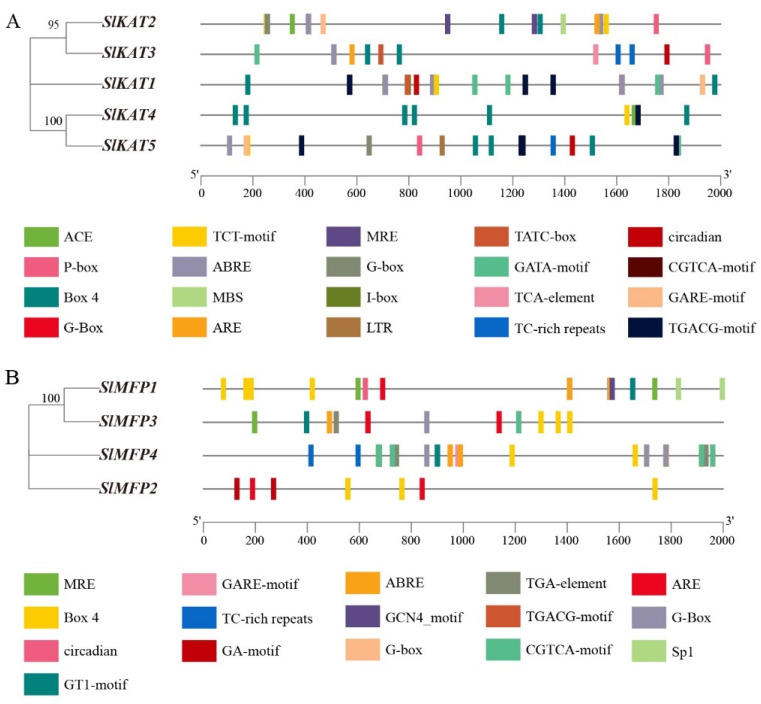
The distribution of *cis*-acting elements in the tomato *SlKAT* and *SlMFP* genes. Different colors represent different *cis*-acting elements. (**A**) The distribution of *cis*-acting elements in the *KAT* family genes. (**B**) The distribution of *cis*-acting elements in the *MFP* family genes.

**Figure 6 ijms-25-02273-f006:**
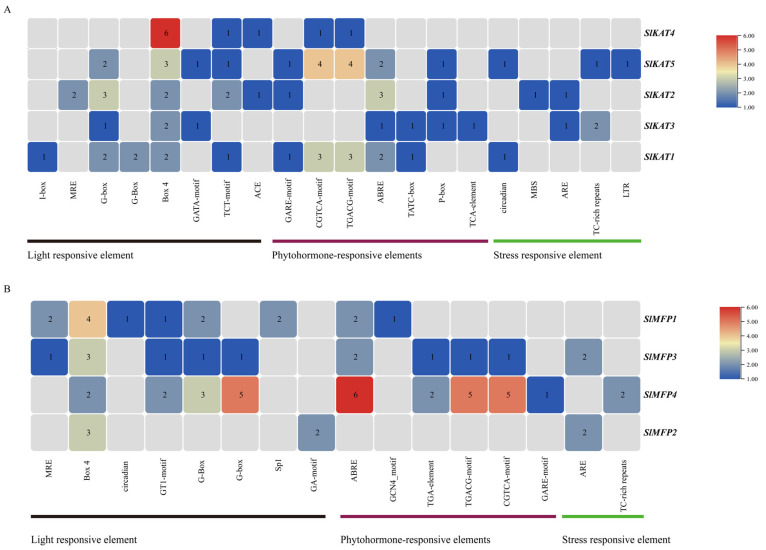
Analysis of the number of *cis*-acting elements of the tomato *SlKAT* and *SlMFP* genes. (**A**) Analysis of the number of *cis*-acting elements of the *KAT* gene. (**B**) Analysis of the number of *cis*-acting elements of the *MFP* gene.

**Figure 7 ijms-25-02273-f007:**
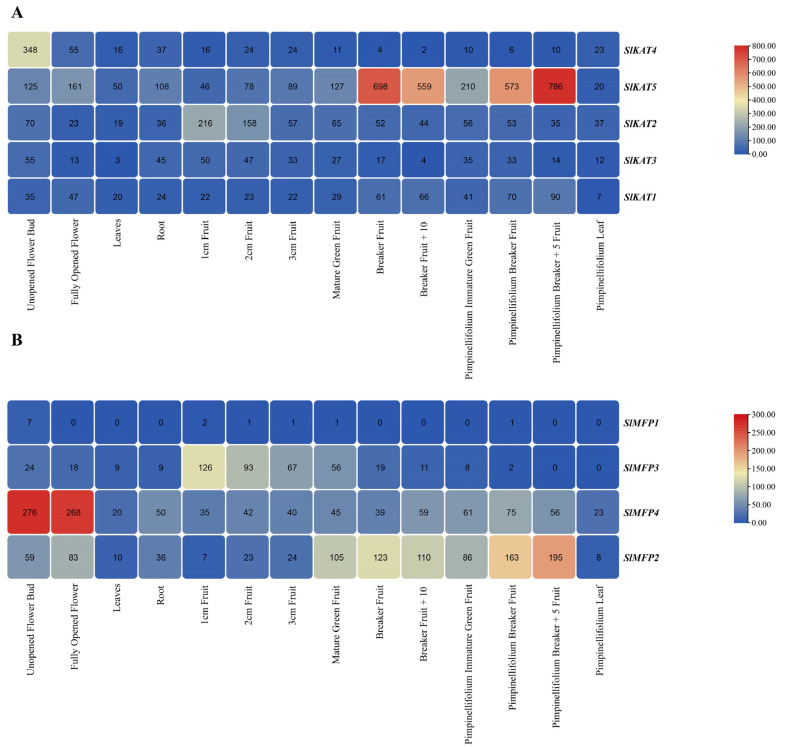
Expression patterns of tomato *SlKAT* and *SlMFP* in different tissues. The color scale represents the fold change normalized by the log2-transformed data. Different numbers represent the level of significance. Heatmaps are shown in blue/yellow/red for low/medium/high expression, respectively. (**A**) Expression of tomato *SlKAT* in different tissues of tomato plants. (**B**) Expression of tomato *SlMFP* in different tissues of tomato plants.

**Figure 8 ijms-25-02273-f008:**
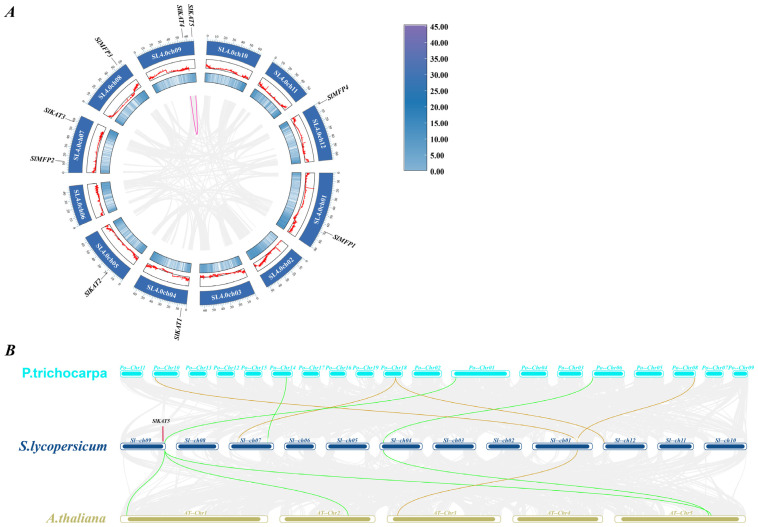
Gene duplication and collinearity analysis of *KAT* and *MFP* genes. (**A**) Covariance analysis of *SlKATs* with *SlMFPs*. The blue bars represent chromosomes, the gray lines represent all co-lined genes in tomato, and the pink lines indicate *KAT* and *MFP* co-lined genes. The color scale represents the fold change normalized by the log2-transformed data. (**B**) Covariance of *P. tomentosa*, tomato, and *A. thaliana* genes. Yellow lines indicate *KAT* genes and green lines indicate *MFP* genes. Gray lines in the background indicate collinear blocks within the *P. tomentosa*, tomato, and *A. thaliana* genes.

**Figure 9 ijms-25-02273-f009:**
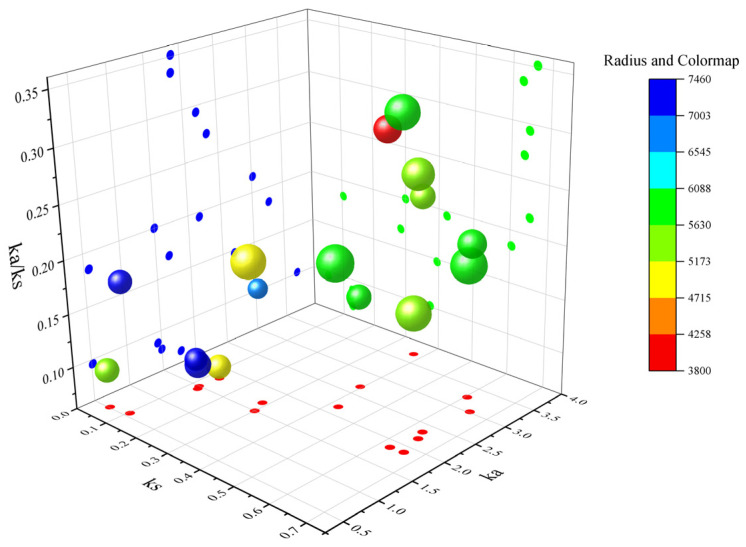
Evolutionary selection pressure analysis of the *SlKAT* and *SlMFP* homologous gene pairs. The *X*-axis represents the Ka value, the *Y*-axis represents the Ks value, and the *Z*-axis represents the ratio of Ka to Ks. The color scale represents the fold change normalized by the log2-transformed data.

**Figure 10 ijms-25-02273-f010:**
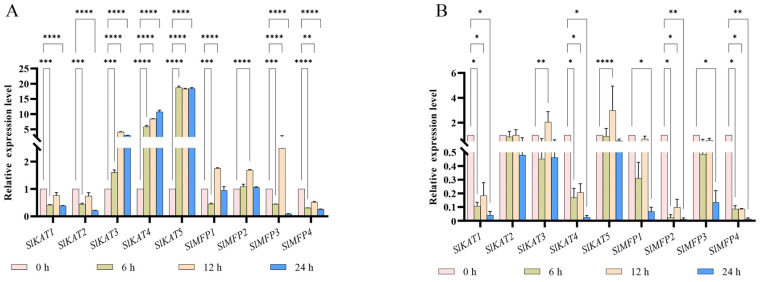
Expression levels of *SlKAT* and *SlMFP* genes under ABA (**A**) and MeJA (**B**)**.** The asterisk (*) indicates that the expression level of the stress group is significantly different from that of the control group (* *p* < 0.05, ** *p* < 0.01, *** *p* < 0.001, and **** *p* < 0.0001, one-way ANOVA, Tukey test). The samples in the 0 h treatment were used as controls.

**Figure 11 ijms-25-02273-f011:**
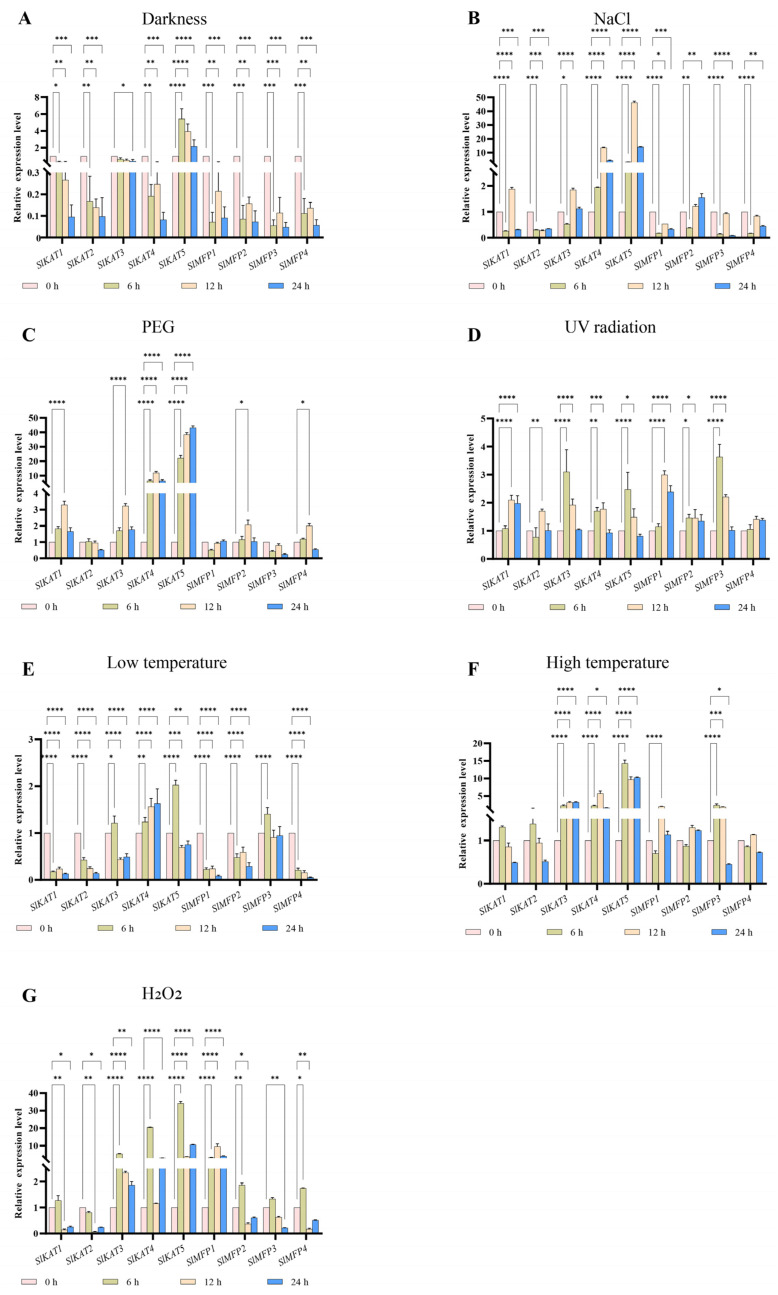
Darkness (**A**), NaCl (**B**), PEG (**C**), UV radiation (**D**), low-temperature (**E**), high-temperature (**F**), and H_2_O_2_ (**G**) treatments. The asterisk (*) indicates the corresponding gene that was significantly up- or down-regulated compared with the 0 h status (* *p* < 0.05, ** *p* < 0.01, *** *p* < 0.001, and **** *p* < 0.0001, one-way ANOVA, Tukey test). The samples in the 0 h treatment were used as controls.

**Table 1 ijms-25-02273-t001:** Genomic information on members of the *SlKAT* and *SlMFP* family genes in tomatoes.

Gene	Gene ID	Gene Locus	ORF (bp)	Amino Acid	Molecular Weight (Da)	pI	GRAVY
*SlKAT1*	Solyc04g015100.3.1. ITAG4.0	Chr04	1245	414	42,901.37	8.87	0.17
*SlKAT2*	Solyc05g017760.4.1. ITAG4.0	Chr05	1146	381	39,128.82	5.73	0.12
*SlKAT3*	Solyc07g045350.4.1. ITAG4.0	Chr07	1215	404	41,312.49	6.47	0.156
*SlKAT4*	Solyc09g061840.4.1. ITAG4.0	Chr09	1446	481	50,588.34	8.67	0.116
*SlKAT5*	Solyc09g091470.3.1. ITAG4.0	Chr09	1398	465	49,024.19	7.97	−0.023
*SlMFP1*	Solyc01g066620.4.1. ITAG4.0	Chr01	1821	606	67,336.01	9.04	0.043
*SlMFP2*	Solyc07g019670.4.1. ITAG4.0	Chr07	2172	723	78,397.7	9.33	0.092
*SlMFP3*	Solyc08g068390.3.1. ITAG4.0	Chr08	2133	710	77,162.22	5.68	0.029
*SlMFP4*	Solyc12g007170.3.1. ITAG4.0	Chr12	2148	715	77,793.88	9.11	0.108

**Table 2 ijms-25-02273-t002:** Detailed information on the 10 conserved motifs of the tomato KAT protein.

Motif	Width (aa)	Motif Sequence
Motif 1	50	SQIDYYEINEAFAVVALANQKLLGLBPEKINVHGGAVSLGHPLGCSGARI
Motif 2	46	GMGVCAENCAERFKITREEQDQYAVQSFERGIAAQESGAFAWEIVP
Motif 3	50	PNTVICTTVNKVCSSGLKATMLAAQSIQAGFNDIVVAGGMESMSNVPKYL
Motif 4	50	VLVSGEKAIKLGLPVIGKIRGFADAAQEPELFGIAPALAIPKAIKSAGLE
Motif 5	29	KANVDPSLVEEVFFGNVLSANLGQAPARQ
Motif 6	29	EARKGSRLGHDSLVDGMLKDGLTDVYNDC
Motif 7	21	TLLGVLRQKDGKFGVAGVCNG
Motif 8	41	EVSGGRGRPSVIVDKDEGLAKFDGAKLRKLRPSFKDTDGTV
Motif 9	15	TAGNASQISDGAAAL
Motif 10	21	MGGFLGSLSSLSATKLGSIAI

**Table 3 ijms-25-02273-t003:** Detailed information on the 10 conserved motifs of the tomato MFP protein.

Motif	Width (aa)	Motif Sequence
Motif 1	50	LDYEDFKDVDMVIEAVIENVPLKQQIFIDIEKVCPPHCILASNTSTIDLN
Motif 2	50	HFFSPAHVMPLLEIVRTEKTSPQVILDLMAVGKAIKKVPVVVGNCTGFAV
Motif 3	49	EMILFPVVNEACRVLDEGIVVRASDLDIASVLGMSFPSYRGGIVFWADT
Motif 4	50	LGCHARIAAPRAQLGLPELSLGVIPGFGGTQRLPRLIGLSKAVDMMMTSK
Motif 5	50	LVHIFFAQRATSKVPNVTDIGLKPRSIKKVAIIGGGLMGSGIATALILSN
Motif 6	49	NRTFFPYSQGAQLLVNLGVDVYRIDTQITEFGLPMGPFQLQDLAGYGVA
Motif 7	41	MSEEGKELGLIDAIVPSDELLKVARRWALDIAERRKPWMRA
Motif 8	50	DRVFKSPLVDLLIKSGRNGKNNGKGYYIYDKGRKPRPDPSVLPIIEESRR
Motif 9	38	VGAGHIYSSLKKWSEIYGNFFKPSKFLEEKAAKGIPLS
Motif 10	41	NMPQHLACJDVIEEGIVHGGYNGJLKEAKVFEDLVLSDTSK

**Table 4 ijms-25-02273-t004:** Secondary structure of KAT and MFP family members in the tomato.

Protein	Alpha Helix (%)	Beta Turn (%)	Random Coil (%)	Distribution of Secondary Structure Elements
SlKAT1	45.65	8.94	31.64	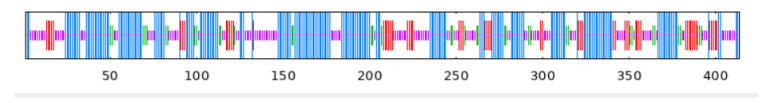
SlKAT2	49.61	9.45	26.77	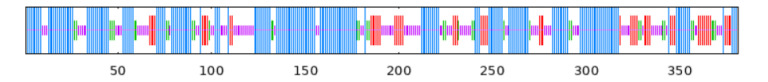
SlKAT3	46.04	8.91	30.20	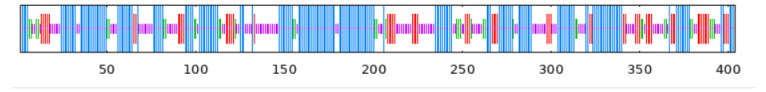
SlKAT4	39.71	8.11	36.17	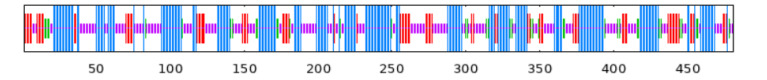
SlKAT5	44.52	7.53	32.69	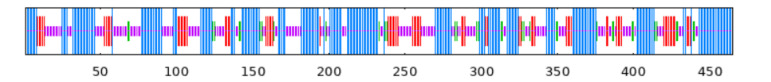
SlMFP1	52.31	5.28	33.83	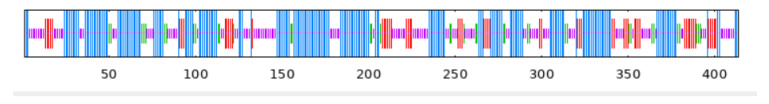
SlMFP2	50.21	7.75	31.67	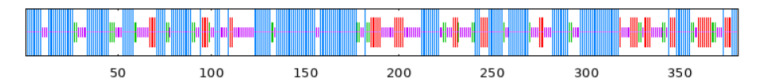
SlMFP3	50.00	7.04	32.82	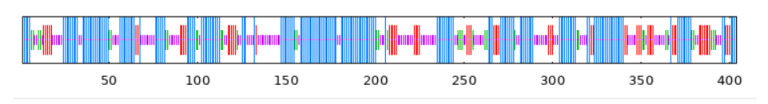
SlMFP4	49.37	6.99	33.85	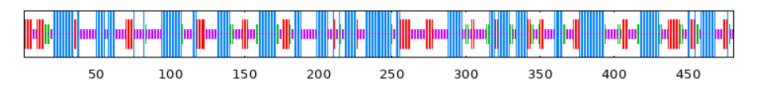

**Table 5 ijms-25-02273-t005:** Prediction of subcellular localization of SlKAT and SlMFP family members in tomatoes.

Protein	Nucleus	Chloroplast	Cytosol	Peroxisome	Plasma Membrane	Endoplasmic Reticulum
SlKAT4	—	4	4	1	—	—
SlKAT5	2.5	1	—	2	1	—
SlKAT2	—	6	7	1	—	—
SlKAT3	—	3	7	3	—	—
SlKAT1	1	9	3	—	—	—
SlMFP1	—	3	3	7	—	—
SlMFP3	1	1	6	1	—	4
SlMFP4	—	1	—	1	8	3
SlMFP2	1	—	1	12	—	—

## Data Availability

All data, tables, and figures in this manuscript are original, and are contained within the article and Appendix A.

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
