# Peer review of "Genome-Wide Identification and Characterization of Tomato Fatty Acid β-Oxidase Family Genes KAT and MFP"

_ijms, 2024, doi:10.3390/ijms25042273_

Round 1

Reviewer 1 Report

Comments and Suggestions for Authors

Dear authors,
To what extent is the information obtained new and to what extent is it a confirmation of other studies? I would also like to ask about the practical aspect of your research: what impact does it have on horticultural practise (now or in the future)? Since the members of the KAT and MFP family play an important role in regulating the development of flower organs and fruit ripening, is it possible to stimulate flowering and fruit set and thus increase yields?
Does a change in light conditions (improvement, deterioration) influence gene expression or dependence? For example, does too intense light (which causes stress in tomatoes) lead to the expression of SlKAT and SlMFP and the production of stress proteins? How does this affect chlorophyll fluorescence/photosynthetic efficiency (and later biomass/yield)? I would appreciate it if the authors would address the practical applications of the research in the discussion, if not now, then in the future.

Comments on the Quality of English Language

The English language is not objectionable, the text is comprehensible and requires little intervention.

Author Response

Dear Editor,

Thanks a lot for having reviewed our manuscript (ijms-2847012). We have revised the manuscript and would like to submit it for your consideration. According to your comments and suggestions, we have made corresponding changes. The revisions have been highlighted in the revised manuscript.

I greatly appreciate both your help and that of the referees concerning improvement to this paper. Below you can find point-to-point responses to Reviewers’ comments. We hope that the revised version of the manuscript is now acceptable for publication in your journal.

I look forward to hearing from you soon.

We would like to express our sincere thanks again to you for the constructive and positive comments.

With best wishes,

Yours sincerely,

Long Li, Chunlei Wang

Response to Reviewer #1

Comment 1:

To what extent is the information obtained new and to what extent is it a confirmation of other studies?

Response:

Thank you for your important questions. Research on the KAT and MFP genes in plants is limited, and there have been no studies on these genes in tomatoes so far. Thus, our research plays a certain role in revealing the impact of the KAT and MFP genes on growth and development of tomatoes. The role of the fatty acid β-oxidation pathway has been studied in plants, and studies of the KAT and MFP genes also exist in Arabidopsis thaliana. Thus, our study could also reinforce the research concerning tomato fatty acid β-oxidation. In detail, in our study, the identification of tomato SlKAT and SlMFP family genes, gene structure, construction of evolutionary trees, chromosomal localization, collinearity analysis, evolutionary pressure selection analysis, and qRT-PCR experiments are among the most recent results of our research. However, the physicochemical properties, secondary structure, and subcellular localization of tomato SlKAT with SlMFP family members were obtained directly from the database (https://web.expasy.org/protparam/, https://wolfpsort.hgc.jp/, https://wolfpsort.hgc.jp/). Meanwhile, the conserved motifs, cis-acting element analyses and tissue expression analyses of tomato SlKAT and SlMFP family members were obtained from databases and then visualized.

Comment 2:

I would also like to ask about the practical aspect of your research: what impact does it have on horticultural practice (now or in the future)? Since the members of the KAT and MFP family play an important role in regulating the development of flower organs and fruit ripening, is it possible to stimulate flowering and fruit set and thus increase yields? Does a change in light conditions (improvement, deterioration) influence gene expression or dependence? For example, does too intense light (which causes stress in tomatoes) lead to the expression of SlKAT and SlMFP and the production of stress proteins? How does this affect chlorophyll fluorescence/photosynthetic efficiency (and later biomass/yield)? I would appreciate it if the authors would address the practical applications of the research in the discussion, if not now, then in the future.

Response:

(i) Thank you for raising this important issue. We would be very happy to discuss the practical implications of our research. Our research has important real and future implications for horticultural practice. Our research contributes to a deeper understanding of plant response mechanisms to abiotic stresses, which is important for improving crop resistance and adaptation. Meanwhile, the role of tomato SlKAT and SlMFP genes in flower organ development and fruit ripening will also provide a theoretical basis for the increase of fruit quality and yield. The corresponding descriptions were added in the revised version of the manuscript. The related statements were as follows:

L482-492: “In addition, ACX4, KAT2 and MFP2 proteins involved in fatty acid β-oxidation are also involved in the ROS1-mediated DNA demethylation process in A. thaliana [51]. What’s more, the DNA demethylation in plants plays a role in many important processes, including fruit ripening and biotic and abiotic stress responses [52]. In our study, the results of tomato tissue expression analysis showed that SlKAT5 genes were more highly expressed in fruits (Fig. 7), while expression of SlKAT4, SlKAT5, and SlMFP1was obviously altered under salt stress treatment (Fig. 10). Thus, our study may provide some theoretical basis for KAT and MFP genes in tomato fruit development and mitigation of salt stress. However, whether KAT and MFP proteins regulate fruit development and salt stress response through DNA demethylation is waiting to be discovered.” was added.

L748-750: “Le, Tuan-Ngoc, Ulrike Schumann, Neil A Smith, Sameer Tiwari, Phil Chi Khang Au, Qian-Hao Zhu, Jennifer M Taylor, Kemal Kazan, Danny J Llewellyn, and Ren Zhang. 2014. 'DNA demethylases target promoter transposable elements to positively regulate stress responsive genes in Arabidopsis', Genome biology, 15: 1-18.” was added.

L751-753: “Wang, Lishuan, Chunlei Wang, Xinye Liu, Jinkui Cheng, Shaofang Li, Jian-Kang Zhu, and Zhizhong Gong. 2019. 'Peroxisomal β-oxidation regulates histone acetylation and DNA methylation in Arabidopsis', Proceedings of the National Academy of Sciences, 116: 10576-85.” was added.

(ii) Thank you for your comments. In our study, SlKAT and SlMFP family members of tomato may play important roles in regulating the development of floral organs and fruit ripening. The effects of MFP family members on floral organ development have been found in previous studies, and KAT family members play an active role in regulating ABA signaling. In addition, ABA is a key hormone for fruit ripening and plays an important role in the process of fruit ripening. The synergistic action of ABA with ethylene is central to the regulation of fruit set. This evidence indirectly demonstrates the critical role of MFP and KAT in regulating the development of floral organs and fruit ripening. However, it is important to note that flowering and fruiting in plants is a complex process regulated by a number of factors, including light, temperature, water, and nutrients. In practice, the plant growth environment and physiological state, as well as the complexity of gene regulation, need to be taken into account in order to better achieve the goal of increasing yield. The statements concerning the possible role of SlKAT and SlMFP on flowering and fruiting and the related practical applications were added in the “Discussion” section of the manuscript. The changes were as follows:

L421-423: “In addition, whether the absence of the SlMFP4 gene will lead to abnormal development of tomato floral organs, thereby affecting plant fertility, requires further in-depth research” was added.

L405-409: “ABA has an important effect on the accumulation of flavonoids and carotenoids during tomato fruit ripening [53]. In addition, the presence of ABA increases fruit firmness by modifying the tomato cell wall [54] AtKAT2 has a positive regulatory effect on ABA signaling [40]. In our study, SlKAT5 were significantly upregulated in response to ABA treatment.” was added.

L754-755: “Fenn, Matthew A., and James J. Giovannoni. 2021. 'Phytohormones in fruit development and maturation', The Plant Journal, 105: 446-58.” was added.

L756-757: “Kou, XiaoHong, JiaQian Zhou, Cai E. Wu, Sen Yang, YeFang Liu, LiPing Chai, and ZhaoHui Xue. 2021. 'The interplay between ABA/ethylene and NAC TFs in tomato fruit ripening: a review', Plant Molecular Biology, 106: 223-38.” was added.

L409-412: “Hence, AtKAT2 and SlKAT5 may share similar function during plant reproductive development” was changed to “Thus, SlKAT5 may take part in plant reproductive development, which may imply a role of SlKAT5 in tomato yield regulation. However, whether SlKAT5 regulates fruit ripening by directly participating in the ABA response needs to be uncovered in the future.”.

L423-427: “It should be noted that the flowering and fruiting of plants is a complex process regulated by various factors such as light, temperature, water, and nutrients. In practical applications, it is necessary to consider the plant's growth environment and physiological status, as well as the complexity of gene regulation, to better achieve the goal of increasing yield” was added.

(iii) Thank you for your insightful questions regarding the influence of light conditions on gene expression, particularly in relation to SlKAT and SlMFP genes, stress protein production, chlorophyll fluorescence, photosynthetic efficiency, and ultimately biomass/yield in tomatoes. The results of the cis-acting element analyses indicate that there are eight light-related acting elements in the tomato SlKAT and SlMFP family members, respectively. In addition, under dark treatment, the expression levels of SlKAT1, SlKAT2, SlKAT4, SlMFP1, SlMFP2, SlMFP3, and SlMFP4 genes were down-regulated. However, the expression level of SlKAT3 was not significantly changed, and the expression level of SlKAT5 was up-regulated by darkness. The down-regulation of SlKAT1, SlKAT2, SlKAT4, SlMFP1, SlMFP2, and SlMFP3 genes may imply that these genes play an important regulatory role under light conditions, while their expressions are suppressed under dark conditions. Thus, these genes may be involved in plant response to light. However, our research and previous studies have not evaluated the impact of plant KAT and MFP members on chlorophyll fluorescence/ photosynthetic efficiency. Thus, the mechanism of tomato SlKAT and SlMFP in regulating chlorophyll fluorescence/ photosynthetic efficiency is waiting to be discovered. The corresponding descriptions were added in the revised version of the manuscript. The related statements were as follows:

L492-498: “It’s well-known that understanding the intricate relationship between light conditions, gene expression, and physiological responses in plants is essential for optimizing growth conditions and enhancing crop productivity. By investigating these interactions, we can gain valuable insights into the molecular mechanisms underlying plant responses to environmental stimuli, ultimately contributing to strategies for improving biomass and yield in agricultural settings. Thus, the mechanism by which tomato SlKAT and SlMFP respond to light may be important for fruit yield and needs to be further uncovered.” was added.

Reviewer 2 Report

Comments and Suggestions for Authors

Review of the manuscript entitled "Genome-wide identification and characterization of tomato fatty acid β-oxidase family genes KAT and MFP" by Li et al. I find the research interesting and valuable. I recommend that the paper be accepted for publication with major revision. The inaccuracies indicated below should be corrected.

Specific comments:

1. The abstract should include brief information about the scope of the research work and the research methods used.

2. Keywords: repeat the information contained in the title of the paper. This should be amended.

3. L95-96. This is not the purpose of the study.

4. L222-235. Fig. 7. Please provide a description of this experiment in the methodology of the paper.

5. L290. transcritional levels?

6. L298. transcriptiona levels?

7. Fig. 11 is difficult to read. The graph contains a lot of information, but it is unclear which data points correspond to the control plants. Is the 0h combination the control variant? Please clarify.

8. L535-560. Is Micro-Tom a tomato variety? What are its characteristics compared to other varieties? Please specify (list) what the research variants were. Was the experiment repeated in several series? At what growth stage were the plants tested?

9. L561-571. What was the efficiency of RNA isolation? How was the purity and quantity of RNA verified? In how many replicates were the studies conducted?

10. L570. Primer primer 5.0. Please specify software manufacturer.

11. L569-570. Please present the qPCR primers developed for the present study. What was the efficiency of the designed primer sets? How was the reference gene selected for qPCR analyses?

12. L571. Who is the author of the cited method? Please provide literature citations.

Author Response

Dear Editor,

Thanks a lot for having reviewed our manuscript (ijms-2847012). We have revised the manuscript and would like to submit it for your consideration. According to your comments and suggestions, we have made corresponding changes. The revisions have been highlighted in the revised manuscript.

I greatly appreciate both your help and that of the referees concerning improvement to this paper. Below you can find point-to-point responses to Reviewers’ comments. We hope that the revised version of the manuscript is now acceptable for publication in your journal.

I look forward to hearing from you soon.

We would like to express our sincere thanks again to you for the constructive and positive comments.

With best wishes,

Yours sincerely,

Long Li, Chunlei Wang

Response to Reviewer #2

Comment 1:

The abstract should include brief information about the scope of the research work and the research methods used.

Response:

Thank you very much for your comments. We have carefully revised the descriptions in the section of “Abstract”, and the brief information about the scope of the research work and the research methods were added. The changes were as follows:

L17-21: “Their physicochemical properties, protein secondary structure, subcellular localization, gene structure, phylogeny and collinearity were also analyzed. In addition, conserved motif analysis, evolutionary pressure selection analysis, cis-acting ele-ment analysis, tissue expression profiling and qRT-PCR analysis were conducted within tomato KAT and MFP family members.” was added.

L31-33: “These results will provide a basis for studying the functions of the KAT and MFP family genes in tomato” was changed to “These results provide a basis for the involvement of the SlKAT and SlMFP genes in tomato floral organ development and abiotic stress response, which laid a foundation for future functional study of SlKAT and SlMFP in tomato”.

Comment 2:

Keywords: repeat the information contained in the title of the paper. This should be amended.

Response:

Thank you for your comments. We are very sorry for the mistake in the part of “Keyword”, we have changed the information of keywords according to your suggestions. The changes were as follows:

L34-35: KAT; MFP; Fatty acid β-oxidation; Tomato” was changed to “multifunctional protein; 3-ketolipoyl-CoA thiolase; Fatty acid metabolism; Abiotic stresses; Tissue Expression Profiling”.

Comment 3:

L95-96. This is not the purpose of the study.

Response:

Thank you very much for your kind comments. According to your suggestions, we have modified the description of the purpose of the study. The related statements were as follows:

L102-104: “This study will provide valuable information for the further exploration of KAT and MFP members in plants” was changed to “Our study extends the information of KAT and MFP family genes in plants and suggests potential roles of KAT and MFP family members in phytohormone and abiotic stress responses”.

Comment 4:

L222-235. Fig. 7. Please provide a description of this experiment in the methodology of the paper.

Response:

Thank you for your valuable comments. We have revised the description of the “Materials and Methods” section. The changes were as follows:

L556-557: “Tissue expression information for the tomato SlKAT and SlMFP genes was obtained from the eFP (http://bar.utoronto.ca/efp/cgi-bin/efpWeb.cgi) database” was added.

L559-562: “The data were organized and the expression patterns of SlKAT and SlMFP in different tissues were mapped using TBtools” was changed to “We obtained tissue-specific expression analysis information from the eFP database using the tomato SlKAT and SlMFP gene IDs. We organized data on the expression of SlKAT and SlMFP in different tissues of tomato and then used TBtools to map the expression patterns of SlKAT and SlMFP in different tissues”.

Comment 5:

L290. transcritional levels?

Response:

Thank you very much for your comments and we would apologize for this error. We have made changes based on your suggestions in the revised manuscript. The change was as follows:

L297: “transcritional levels” was changed to “transcriptional levels”.

Comment 6:

L298. transcriptiona levels?

Response:

Thank you very much for your comments and we are very sorry for the mistake. We have made changes based on your suggestions in the revised manuscript. The change was as follows:  

L304: “transcriptiona levels” was changed to “transcriptional levels”.

Comment 7:

Fig. 11 is difficult to read. The graph contains a lot of information, but it is unclear which data points correspond to the control plants. Is the 0h combination the control variant? Please clarify.

Response:

Thank you very much for your valuable comments and specific enquiries about Fig. 11. We hereby provide the necessary clarifications and additional information to improve the readability of the figure and to clarify the corresponding data points. The data combination labelled "0h" in Fig. 11 was used as a control variant in our experiments. Plant materials in the "0h" treatment were intended to provide a baseline against which expression could be compared with other time points under each treatment condition. We have made changes in Fig. 11 and its figure legend. Also, in the “Materials and Methods” section of the paper, the detailed information about the control group setup and experimental design were added in “4.9.”. The related changes were as follows:

L332-334: “The asterisk (*) indicates that the expression level of the stress group is significantly different from that of the control group (*p <0.05, **p < 0.01, ***p < 0.001, ****p < 0.0001, one-way ANOVA, Tukey test)” were changed to “The asterisk (*) indicates the corresponding gene significantly up- or down-regulated compared with the 0 h statuses (*p <0.05, **p < 0.01, ***p < 0.001, ****p < 0.0001, one-way ANOVA, Tukey test). The samples in 0 h treatment was used as controls.”

L620-621: “The relative expression values of each gene under each treatment at different treated times were calculated via comparing with that at 0 h” was added.

Comment 8:

L535-560. Is Micro-Tom a tomato variety? What are its characteristics compared to other varieties?

Response:

Thank you for your valuable comments. Taking your suggestions into account, we have added a detailed description for tomato ‘Micro-Tom’. Plant seeds of the tomato (Solanum lycopersicum L.) cultivar ‘Micro-Tom’ were originally supplied by A Levy in Israel. Based on its compact habit, ‘Micro-Tom’, a dwarf cultivar of tomato (Solanum lycopersicum L.), has been proposed as a preferred variety to carry out molecular research in tomato. The related statements were as follows:

L573-575: Micro-Tom is a tomato variety widely used in scientific research. Micro-Tom is characterized by its small size, short growth cycle, small fruits, and high degree of self-fertilization [50]” was added.

L747: “Marti, E. 2006. 'Genetic and physiological characterization of tomato cv. Micro-Tom', Journal of Experimental Botany, 57: 2037-47.” was added.

Comment 9:

Please specify (list) what the research variants were. Was the experiment repeated in several series? At what growth stage were the plants tested?

Response:

Thank you for your valuable comments. Taking your suggestions into account, we have added the description of the experimental treatments. Tomato ‘Micro-Tom’ was used as the plant study subject and the plants under 0 h treatments were control plants were. Our study involved several treatment variants including ABA, MeJA, darkness, NaCl, PEG, UV, cold, heat and H2O2 treatments. The experimental groups were also treated for different durations, including 6 h, 12 h and 24 h. Each treatment of each treated time contained three biological replicates, and each replicate consisted of 8 seedlings. The related statements were as follows:

L599-603: “Each treatment of each treated time contained three biological replicates, and each replicate consisted of 8 seedlings. All plants were tested at the seedling stage to ensure that treatment effects were assessed at the same physiological stage. The experimental groups varied in their treatments and treatment times” was added.

L583-585: “After 21 days, the seedlings of uniform size were selected for subsequent treatments” was changed to “Plant material was grown for a fortnight and then transplanted from soil to nutrient solution for hydroponics. Then, 21-d-old seedlings with uniform size were used for different treatments”.

Comment 10:

L561-571. What was the efficiency of RNA isolation? How was the purity and quantity of RNA verified?

Response:

Thank you for your inquiry regarding the efficiency of RNA isolation and the verification of RNA purity and quantity in our studies. TRIzol maintains the integrity of the RNA when breaking up and lysing the cells, making it useful for the purification of RNA and the production of standardized RNA. We verified the purity and quantity of RNA using a Pultton P100+ ultra-micro spectrophotometer (Wuzhou Dongfang, Beijing, China). We selected RNA samples with A260/A280 ratios between 2.0 and 2.1 for subsequent experiments. Meanwhile, the concentrations of RNA were all greater than 500 ng/μL. In addition, the corresponding references were added in the revised manuscript. More detailed information please see the revised manuscript. The related changes were as follows:

L605-609: “Total RNA was extracted from the samples using TRIzol reagent (Invitrogen, Carlsbad, CA, USA).” was changed to “Total RNA was extracted from the samples using TRIzol reagent (Invitrogen, Carlsbad, CA, USA) [35,36]. The purity and concentration of RNA was then examined by a Pultton P100+ ultra-micro spectrophotometer (Wuzhou Dongfang, Beijing, China). The A260/A280 ratios of RNA samples between 2.0 and 2.1 were chosen for the subsequent experiments”.

Comment 11:

In how many replicates were the studies conducted?

Response:

Thank you for your inquiry regarding the efficiency of the number of replicates conducted in our studies. Three independent replications were performed for all experiments in our study, and each replication contain 8 plants. The related changes were as follows:

L621-623: “All experiments in our study were repeated three times independently to ensure the reliability and statistical significance of the results” was added.

Comment 12:

L570. Primer primer 5.0. Please specify software manufacturer.

Response:

Thank you for your review comments. We are very sorry that we have misspelled the name of the software. We have corrected this mistake and added the information of the specify software manufacturer in the revised manuscript. The changes were as follows:

L615-616:“Primer primer 5.0” was changed to “Primer Premier 5.0(Premier Biosoft, USA)”

Comment 13:

L569-570. Please present the qPCR primers developed for the present study.

Response:

Thank you very much for your comments. Please excuse our oversight and we have added the primer detailed information in Table S1. The corresponding supplementary file has been added in the revised files. In addition, the corresponding description was revised in the revised version of the manuscript. The changes were as follows:

L618:“Table 1” was changed to “Table S1”.

Comment 14:

What was the efficiency of the designed primer sets? How was the reference gene selected for qPCR analyses? 

Response: Thank you very much for your careful review and the specific questions you asked. The following is our description of primer efficiency and reference gene selection. We assessed the efficiency of the designed primer sets by constructing standard curves. Specifically, we used a dilution series of template DNA for qPCR and then calculated the amplification efficiency of each primer pair. The amplification efficiencies of all designed primer pairs for our experiments ranged from 90% to 110%, ensuring good amplification efficiency and specificity of the primers. Primer lengths were between 18 bp-30 bp with a GC% ≤ 60. We chose SlActin (NC015447.3) as the reference gene because it showed the most stable expression level among all the samples tested in our previous study. This choice ensured the accuracy and reliability of our qPCR data. In addition, we have added the related reference information to supplement our note in the revised version of the manuscript. The related changes were as follows:

L616-617:“and the internal reference was SlActin (NC015447.3) as was shown in Table S1” was changed to “and the internal reference was SlActin (NC015447.3) as was shown in Table S1, which showed stable expression level among all the samples tested in our previous study [22]”.

Comment 15:

L571. Who is the author of the cited method? Please provide literature citations.

Response:

Thank you for your comments. We have added the information of the corresponding reference based on your suggestions. The changes were as follows:

L618-620:“The 2-∆∆CT calculation method was used to quantify the relative expression of each gene” was changed to “The 2-∆∆CT calculation method was used to quantify the relative expression of each gene as described in Schmittgen, T. D [49]”.

L746:“Schmittgen, T. D., and K. J. Livak. 2008. 'Analyzing real-time PCR data by the comparative C(T) method', Nat Protoc, 3: 1101-8.” was added.

Round 2

Reviewer 2 Report

Comments and Suggestions for Authors

The paper has been revised. All comments have been clarified. I recommend acceptance of the work in its present form.